# Preferences of the Facade Composition in the Context of Its Regularity and Irregularity

Michał Malewczyk *, Antoni Taraszkiewicz and Piotr Czyż

Faculty of Architecture, Gdańsk University of Technology, 80-233 Gdańsk, Poland; antarasz@pg.edu.pl (A.T.); pioczyz@pg.edu.pl (P.C.)
* Correspondence: michal.malewczyk@pg.edu.pl

**Abstract:** The aim of this study is to determine the preferences of Polish society towards building facades depending on the degree of the composition regularity of the facade elements. The subject matter is inspired by the authors' observations in relation to the current architectural trends. The purposefulness of the conducted research results from several issues. Firstly, the reports of psychology and neurosciences clearly indicate the universality of certain preferences towards visual attributes of objects (e.g., in relation to abstract symmetric patterns), resulting from biological conditions. Secondly, residential, multi-family architecture is by definition designed for a wide group of anonymous users whose expectations must be met. One of the dimensions of the above-mentioned expectations is the visual dimension, partially dependent on the composition of the facade. In the course of the conducted research, it is shown that facades with a regular composition are assessed as more attractive than those with irregular compositions. Moreover, irregular facades evoked a negative effect of a significantly greater force than the positive effect in the case of regular facades. The above-described discoveries shed, in the authors' opinion, a completely new light on the contemporary work of architects. It is extremely important to adapt the visual dimension of architecture to the expectations of its recipients, while taking care of its values and quality as a field of art.

**Keywords:** composition; aesthetics; multi-family housing; Poland; preferences

## 1. Introduction

The main purpose of this study is to determine the aesthetic preferences of Poles depending on the degree of regularity in the composition of building facades. An additional goal is to identify these preferences depending on the type of composition. Aesthetics is naturally a very broad and complex concept, but in relation to this study, it should be considered as a visual dimension of the external part of architecture. The research further narrows the visual dimension of architecture, which is also a broad and complex phenomenon to the composition of window openings (an essential architectural element on most multi-family building facades). Such a limitation allows to limit the conclusions from the study only to the composition of window openings. These should be considered both as a pattern (according to Alexander's theory [1]) and as a component of architecture as a multi-layer composition [2] with an influence on the aesthetics of the whole building.

Therefore, a questionnaire survey was conducted among 109 people. Statistical analyses of the obtained results were used to formulate answers to research questions and conclusions. The significance of the conducted research resulted from the utilitarian nature of residential, multi-family architecture and the resulting universality. Multifamily housing constitutes the main part of the urban fabric and it is extremely important that its aesthetics meet the expectations of modern society. It is impossible to fully meet these expectations without knowing the corresponding statistical data. Although this study refers to the preferences of Poles, and, therefore, its scope is local, it refers to common preferences applicable in the analysed group. It should also be noted that the presented research method is a

universal one and is a starting point for global research and for determining the preferences of the general public.

One of the key issues for this study is the concept of aesthetics. Naturally, this is a very broadly understood term [3]. Aesthetics itself is a branch of philosophy; however, aesthetics understood as the visual dimension of an object (e.g., works of art) also depends on other philosophical trends and concepts. Nowadays, we can observe the emergence of voices suggesting the need to redefine the directions in which the aesthetics of architecture should follow. In the opinion of the authors, special attention should be paid to trends relating to the aesthetics of the everyday [4], an example of which may be the Super Normal [5] initiative. Roger Scrouton [6], among others, draws attention to the importance of creating the aesthetics of architecture in the context of its impact on everyday life. Everyday architecture is also treated as a counterpoint to a world engulfed in consumerism and the necessity to follow fashion [7], a counterpoint to architecture defined by Tom Dyckhoff as "wowhaus" architecture [8]. The need to direct the way of creating architecture to its user is also indicated by Jeremy Till, who called architecture a frame for life [9]. It is very important to fill the gap between architectural monuments and people, as Till also states [9], but in the authors opinion it is not possible without obtaining the knowledge of the real expectations and preferences of ordinary users of architecture.

Another important issue for these considerations is the growing interest in irregularity in the context of shaping the visual dimension of architecture. This tendency is evidenced by many international projects, but the most important for these considerations are objects from Poland. It is worth paying attention to the Polish multi-family architecture, realized in 2011–2021 and, at the same time, nominated for the Mies van der Rohe award. During this period, seven multi-family projects were created (Figure 1) which were nominated for this prestigious award and, therefore, are examples of architecture of above-average quality. These projects inspire other architects and, above all, set trends in design and also reflect the current trends in this segment of architecture. The analysis of the facades of these objects shows that in the case of four (out of seven) implementations, we deal with irregular compositions and with irregular (in terms of texture or colour) facade materials. The two embodiments also operate in an irregular form. It should also be noted that there is a certain irregularity (formal, compositional or material) in each of the realizations. The results of the above analyses are presented in Table 1. In the opinion of the authors, it can, therefore, be concluded that we are dealing with a clear tendency related to the shaping of the contemporary aesthetics of Polish multi-family architecture.

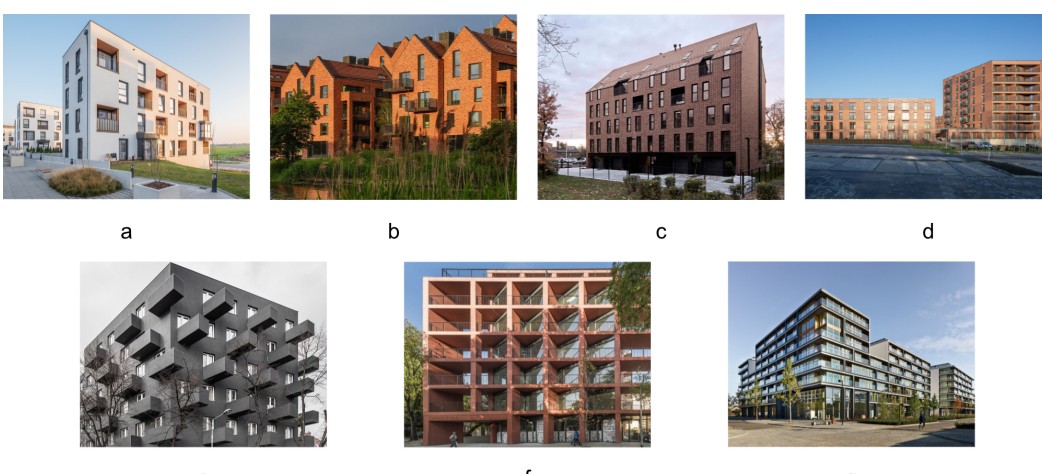

**Figure 1.** Polish residential, multi-family projects nominated for the Mies van der Rohe for 2011–2021: (**a**) Atmosfera real estate; (**b**) Riverview real estate; (**c**) Nowy Werdon building; (**d**) Nowy Nikiszowiec real estate; (**e**) Unikato building; (**f**) Sprzeczna 4 building; (**g**) 19 Dzielnica real estate.

**Table 1.** List of irregularities in relation to the analysed housing developments.

| Project Name | Designer | a | b | c |
|---|---|---|---|---|
| Atmosfera | arch_it | | X | |
| Riverview | APA Wojciechowski | | X | X |
| Nowy Werdon | Biuro Projektowe Maleccy | | X | X |
| Nowy Nikiszowiec | 22ARCHITEKCI | | | X |
| Unikato | KWK PROMES | X | | X |
| Sprzeczna 4 | BBGK | X | | X |
| 19 Dzielnica | JEMS Architekci | | X | |

(a) Irregularity of form. (b) Irregularity of composition. (c) Irregularity of material.

At the same time, due to the utility nature of multi-family housing, a question should be asked whether this direction is in line with the users' expectations. Doubts about the purposefulness of this aesthetic tendency arise from the results of research carried out in many different fields. Firstly, there are many studies which, based on quantitative research, indicate a clear difference between the aesthetic judgments determined by architects and non-architects [10]. Secondly, research clearly shows people's preference for objects and systems with regular features. One should pay attention to the discoveries determined in 2014 by a group of scientists—Pecchinenda, Bertamini, Makin and Ruta [11]. The results of four experiments clearly indicated a tendency to choose patterns or symbols that are symmetrical to asymmetrical. Equally important are the discoveries related to the preferences for fractal-based patterns, which are highly ordered and regular structures. Such patterns are considered to be much more aesthetic than those not based on such structures [12–14]. The same regularity was observed in relation to abstract regular patterns, which, in the research of the Bertamini, Makin and Rampone team, were associated with positively marked words. The associations with irregular patterns were opposite [15]. Thirdly, too much visual variety in the built environment can simply result in spatial chaos.

On the other hand, there are many reasons to move away from extremely repetitive facades. Reports of videoecology clearly indicate the problem of the disappearance of gray brain cells caused by exposure to environments composed of the same, repeatable elements [16,17]. Visually diverse environments also evoke much more positive reactions in recipients [18]. Therefore, it can be assumed that the clear regularity and repeatability in the case of architecture does not reflect the aesthetic needs of a human being, as the results of research by psychologists and neuropsychologists suggest. This could also suggest that we should design irregular compositions that break the monotony and predictability of regular ones. However, in the opinion of the authors, it is not possible to resolve the aesthetic dispute between regularity and irregularity without examining the actual preferences.

An extremely important issue for these considerations is also the issue of universal, biological determinants that govern the processes of perception, determining aesthetic judgments and experiencing beauty [19]. Despite the relatively short tradition of experimental aesthetics or neuroscience, attention should be paid to three publications from 2004 by Vartanian and Goel [20], Kawabata and Zeki [21] and Cela-Conde et al. [22]. All the above-mentioned teams of researchers, thanks to the technology enabling neuroimaging (fMRI or MRI), clearly indicated that when the recipient contacted an object generally considered as beautiful, the reward system was activated in the brain. The results of these studies also highlight the lack of a single structure in the brain responsible for making aesthetic judgments, which are rather the result of more complex processes. Naturally, neurosciences clearly flatten the character of aesthetic experience and may also only provide empirical confirmation of intuitive judgments about beauty [23]. Nevertheless, this approach proves the possibility of describing universal determinants of beauty, common to the general public and independent of the individual's particular characteristics.

Determining the aesthetic preferences regarding the composition of building facades in the context of the degree of their regularity is important as long as they strengthen the designers' awareness of how to shape the space. They can primarily help to determine

the directions in which aesthetics should follow in order to meet the expectations of its recipients. They can also help in determining the directions in which architecture should go in order to simultaneously meet the needs and expectations of its users and, at the same time, to not cease being art. The implementation of this task, however, is the role of architects, whose knowledge should be as broad as possible. The results and conclusions of this study can also be a very powerful planning tool. The use of a specific type of composition in a given area could help with building spatial order, but also with breaking the aesthetic monotony or be an architectural reinforcement of an urban dominant.

## 2. Materials and Method

The aim of the study was to determine the preferences of Poles towards building facades in the context of their regularity. The study was conducted in three stages. The first step was to prepare the research material; then, an internet survey was conducted. The last stage was to carry out statistical analyses of the collected results and to formulate conclusions.

Naturally, each architectural object is a complex being, but each of its components influences the perception of the whole. This approach to the multi-element nature of architecture is reflected in the pattern language proposed by Christopher Alexander [1]. According to the authors, one of the most important patterns is the composition of architectural elements, such as window openings, balconies or elevation panels, because the way the same elements are arranged on the same facade causes its perception to be completely different. The composition also determines the complexity of the facade and the degree of its order [24]. These two determinants were reflected in the parametric method of assessing the aesthetics of an object, proposed by Birkhoff [25] and refined by Eysenck [26]. The composition, therefore, has a significant impact on the perception of the aesthetics of the entire building. A key aspect of the study was to isolate the composition as the test item so as to discover pure preferences for themselves.

### 2.1. Research Questions

1. Is there a relationship between the degree of regularity in facade compositions and aesthetic preferences?
2. Are regular compositions considered more aesthetic than those with a greater degree of irregularity?
3. Is there a relationship between the type of composition and aesthetic preferences?
4. What kind of composition is preferred the most or the least?

### 2.2. Research Material

Five visualizations of the building's facades were prepared for the study. The limitation of the number of stimuli presented during the study resulted from the processing capacity of the human brain. The maximum number of items stored in working memory is between five and nine [27]. Comparing stimuli (which was the task of the respondents during the study) requires storing them in working memory. Therefore, limiting number of stimuli to five minimized the impact of limited human cognitive abilities on the final test result.

The stimuli were digitally generated to maximize control over their final image. The only variable between the stimuli was the composition of the window openings. The window openings were presented as the simplest and no other elements were placed on the facade, so that, according to the theory of multilayer composition [2], the compositions of many layers doid not overlap and did not affect the perception of the whole. The basis for each of the compositions was the elevation (Figure 2) with dimensions of $16 \times 10$ m placed on a uniform background, showing a meadow and greenery in the distance. Reducing the building to a flat facade drawing made it possible to eliminate the influence of perspective on the respondents' perception process. Compositions no. 1–5 (Figure 3) were designed as systems of decreasing degree of regularity, with composition no. 1 having the highest and composition no. 5 having the lowest degree of regularity. This effect was achieved

through the use of compositions that differed in terms of type of composition elements (one, two or three types of composition elements) and the method of building vertical composition lines. The composition type indicated tentatively with the letter "X" (Figure 4) meant compositions composed of elements stacked on top of each other at equal horizontal distances (the highest degree of regularity). Compositions tentatively marked with the letter "Y" (Figure 5) were compositions consisting of clearly marked compositional verticals (the same elements one above the other), but the distances between the vertical lines were different (average degree of regularity). The composition type, initially marked with the letter "Z" (Figure 6), also had different distances between the compositional verticals; however, additionally, these lines were not built by the elements arranged in accordance with the axis of symmetry one above the other, but by the side edges of these elements (the lowest level of regularity).

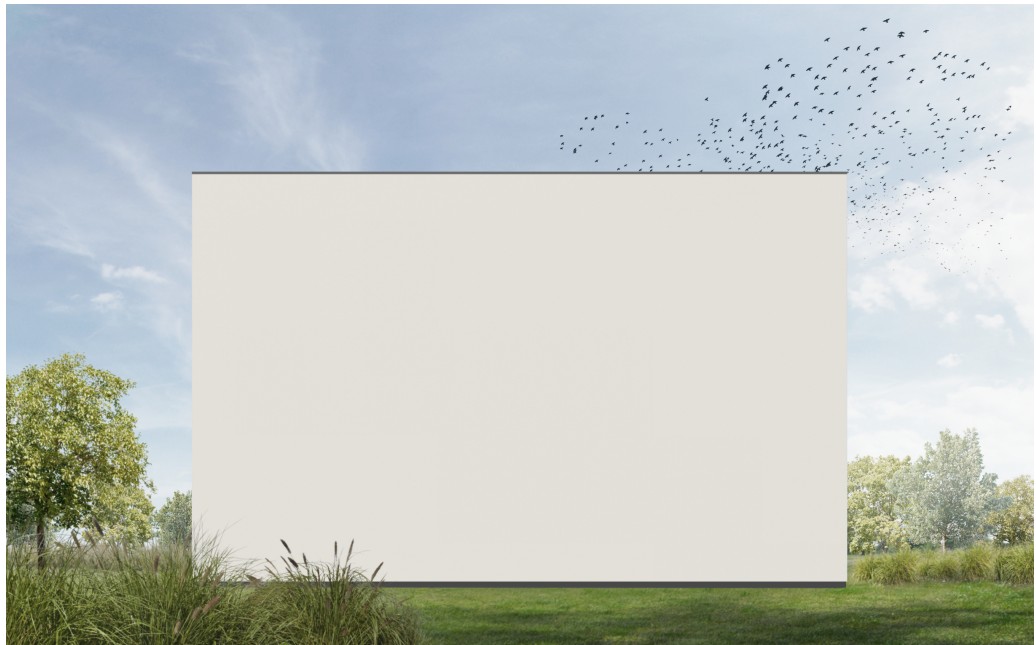

**Figure 2.** The basis for the stimuli for research.

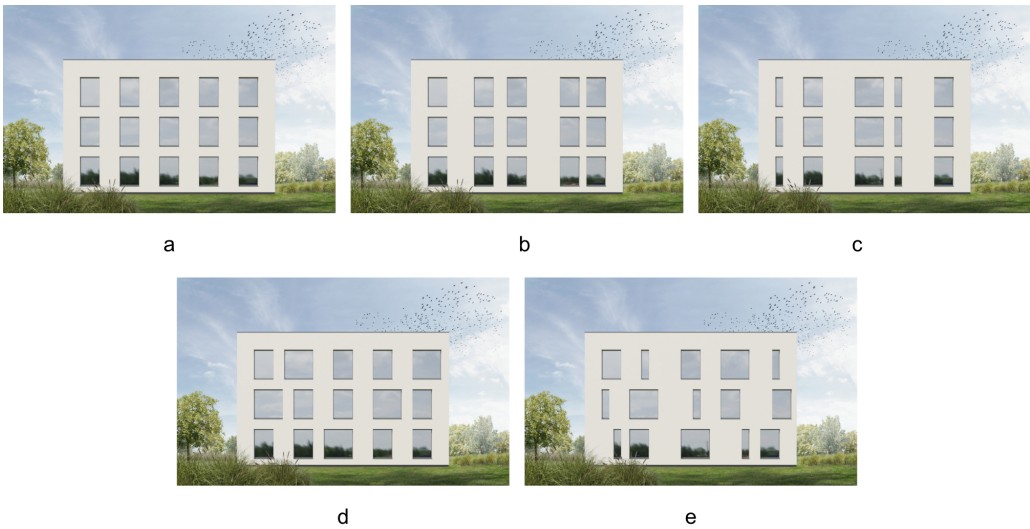

**Figure 3.** The incentives presented in the study: (**a**) composition no. 1; (**b**) composition no. 2; (**c**) composition no. 3; (**d**) composition no. 4; (**e**) composition no. 5.

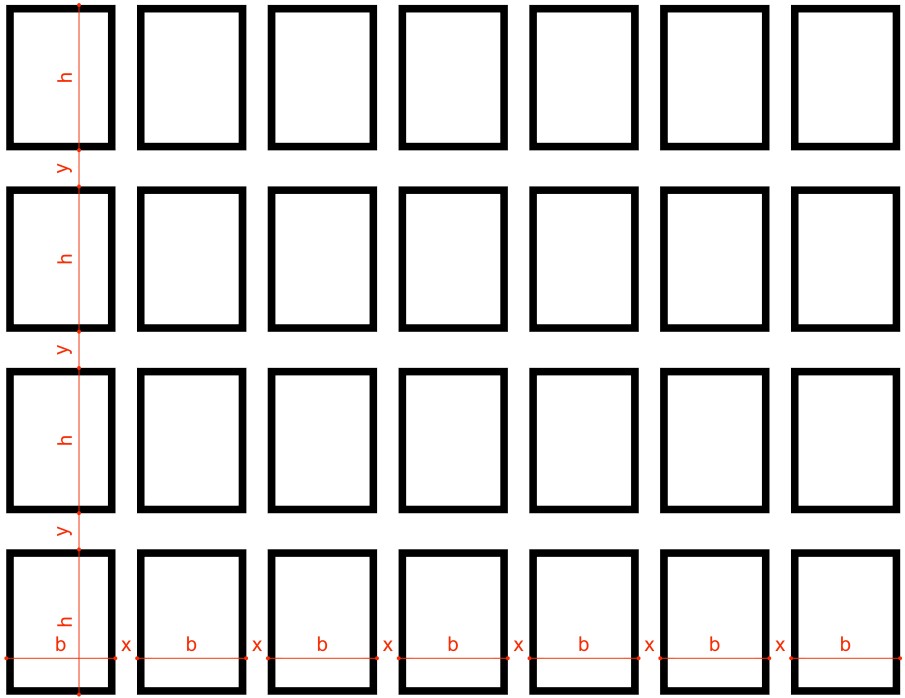

**Figure 4.** Diagram showing the assumptions of an "X" composition.

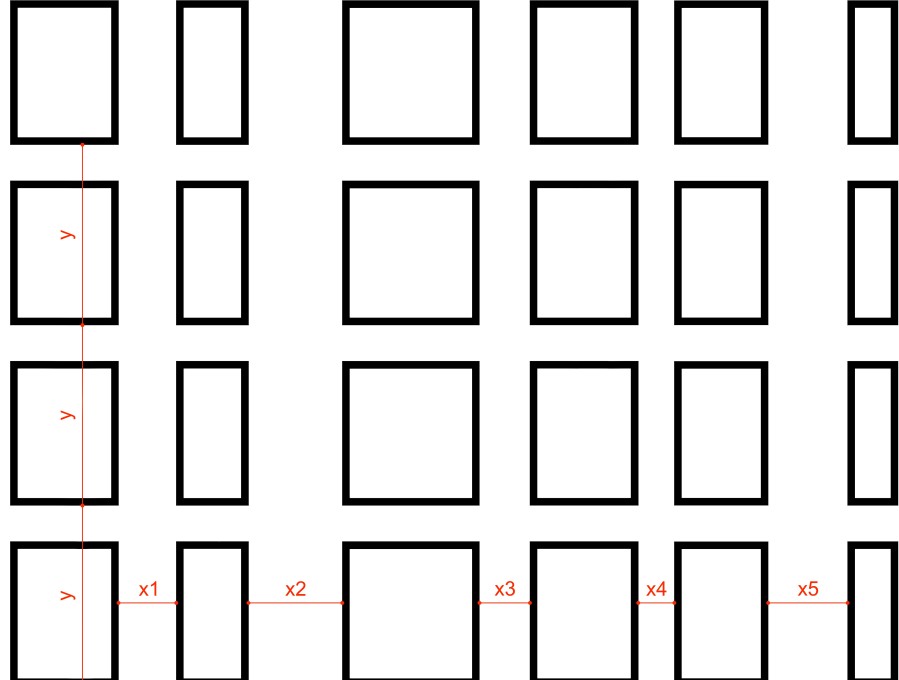

**Figure 5.** Diagram showing the assumptions of a "Y" composition.

Composition no. 1 (Figure 3a) is the composition of type X with the highest degree of regularity. Composition no. 2 (Figure 3b) is an example of a Y-type composition which differed from the first composition only by the horizontal distances between the window openings and used the same compositional elements. Composition no. 3 (Figure 3c) is also an example of a Y-type composition; however, in this case, different widths of the

elements were added, which reduced the degree of regularity of the arrangement with respect to composition no. 2. Composition no. 4 (Figure 3d) is a Z-type composition, based on elements with two different horizontal dimensions. Composition no. 5 (Figure 3e) is also an example of a Z-type composition; however, in this case, elements of three different horizontal dimensions were used. All compositions had 15 elements of equal height arranged in three horizontal lines.

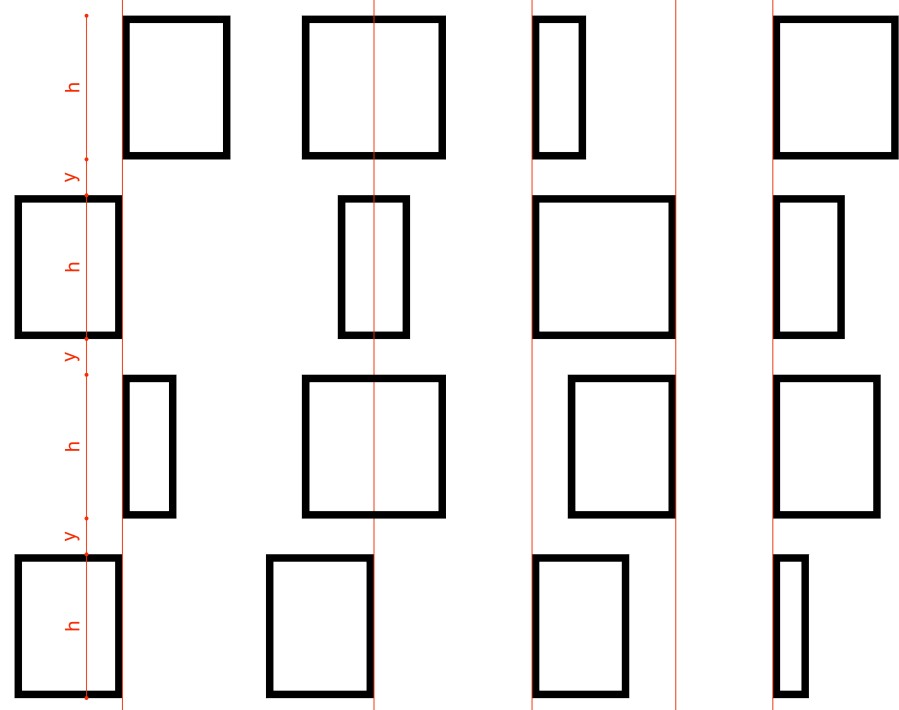

**Figure 6.** Diagram showing the assumptions of a "Z" composition.

The graphics were prepared on the basis of the original 3D model created in Blender, rendered with the Cycles rendering engine and then processed in Affinity Photo.

### 2.3. Variables

The following independent variables were introduced into the study: the degree of regularity, defined by a number from 1 to 5, where 1 is the most regular, and 5 the least regular, and the composition type, marked with the letters X, Y and Z (in accordance with the previously described types of compositions). The dependent variable was the degree of aesthetic preference, indicated by a number from 1 to 5, with 1 being the most preferred and 5 being the least preferred.

### 2.4. Procedure

The performed procedure was based on previous studies [10,28]; however, it used different stimuli, and the rating scale was also limited from 1 to 10 to 1 to 5. On the basis of the prepared research material, an online questionnaire was developed, consisting of two parts. The first part consisted of one task aimed at arranging randomly displayed graphics (described in Section 2.2) from the prettiest (value 1) to the ugliest (value 5). The second part of the questionnaire was not the main element of the study, but it allowed to characterize the research group. This part contained three questions concerning gender, age and education of the respondents, respectively. The survey was conducted from 23 May 2020 to 1 June 2020 and took the form of an online survey. The link to the survey was placed on one of the social networks (Facebook) and shared using a post promotion tool. The message was addressed to people aged 18 to 65 declaring living in Poland.

### 2.5. Characteristics of the Study Participants

The study involved 109 people (N = 109) and met the statistical guidelines for the minimum survey sample for populations larger than 5000 [29]. Margin of error of the research was at the level of 9%, with a confidence level of 95%. Respondents were aged 18 to 62 (M = 31.18, SD = 11.25), of which 75 were women and 34 were men. In total, 73 people had higher education (66.97%), 28 people secondary (25.68%) and 4 people (3.67%) each had primary and vocational education.

## 3. Results

### 3.1. Research Questions 1 and 2

In order to answer research question one, a one-way ANOVA was performed in an intergroup scheme. The dependent variable in these calculations was the degree of regularity (on a scale from one, the most regular, to five, the least regular) to which the presented stimuli belonged. The dependent variable valued the determining aesthetic preferences in relation to stimuli (on a scale from one, the most attractive, to five, the least attractive). A summary of the results obtained during the above-described operation is presented in Table 2.

**Table 2.** ANOVA summary of the variable "degree of aesthetic preference" depending on the degree of regularity of the composition.

| Source | SS | df | MS | F |
|---|---|---|---|---|
| Between groups | 167.76 | 4 | 41.94 | 22.83 ** |
| Inside groups | 890.86 | 485 | 1.84 | |
| Overall | 1058.62 | 489 | | |

** $p < 0.001$.

The analysis of variance turned out to be statistically significant—F (4.489) = 22.833, $p < 0.001$—, which proved the relationship between the degree of regularity of the composition and aesthetic preferences.

Post hoc tests were performed to see if there were differences in aesthetic preferences depending on the particular composition. The result of Levene's test for the variable "aesthetic preferences" turned out to be statistically significant ($p < 0.001$); hence, the assumption of the equality of variance was rejected and Dunnett's T3 test was used. The analyses showed that there was a statistically significant difference in aesthetic preferences between compositions no. 1 and no. 4 and 5, between compositions no. 2 and no. 4 and 5, and between compositions no. 3 and no. 4 and 5.

The above results, together with the results of the descriptive statistics (Table 3), indicated the existence of two groups of compositions and provided answers to research question two. The first group consisted of compositions with a higher degree of regularity, i.e., no. 1 (M = 2.34; SD = 1.428) , no. 2 (M = 2.45; SD = 1.202) and no. 3 (M = 2.64; SD = 1.270), which did not differ statistically significantly in terms of the degree of aesthetic preferences. The second group consisted of compositions with a lower degree of regularity, i.e., no. 4 (M = 3.62; SD = 1.240) and no. 5 (M = 3.68; SD = 1.596), which also did not differ statistically significantly from each other in terms of the degree of aesthetic preferences. In contrast, the compositions of the first group (i.e., no. 1, 2 and 3) were statistically significantly more preferred than the compositions of the second group (i.e., no. 4 and 5).

The data collected during the survey were also subjected to a statistical correlation analysis. A statistically significant, weak, positive correlation was found between the degree of regularity and the degree of aesthetic preference: r = 0.372; $p = 0.01$. Therefore, the higher the degree of regularity in the composition, the higher the degree of aesthetic preference, which was consistent with the analyses carried out earlier on research question two.

However, attention should be paid to the graph (Figure 7) of the distribution of the frequency of assessments of the degree of aesthetic preference depending on the degree

of the regularity of the composition. Although the mean values of the degree of aesthetic preference for the compositions with the degree of regularity 1, 2 and 3 were similar (which was confirmed by the post hoc tests carried out), the response frequency distribution graphs for these compositions clearly differed from each other. Composition no. 1 was significantly more frequently rated as the most aesthetic (37.76% of the response) than compositions no. 2 (21.35% of the response) and no. 3 (16.33% of the response). On the other hand, composition no. 5 was rated as the least visually attractive by 49.49% of the respondents, i.e., 11.73 percentage points more than the most positive grades for composition no. 1. In view of the above, it could be assumed that the irregular compositions elicited negative reactions more strongly than the regular compositions elicited the positive reactions.

**Table 3.** Descriptive statistics for the dependent variable "Degree of Aesthetic Preference" for composition.

| Function | No. 1 | No. 2 | No. 3 | No. 4 | No. 5 |
|---|---|---|---|---|---|
| Standard deviation | 1.428 | 1.202 | 1.270 | 1.240 | 1.596 |
| Standard error | 0.144 | 0.121 | 0.128 | 0.125 | 0.161 |
| The minimum value | 2.05 | 2.21 | 2.39 | 3.37 | 3.36 |
| Mean * | 2.34 | 2.45 | 2.64 | 3.62 | 3.68 |
| The maximum value | 2.62 | 2.69 | 2.90 | 3.87 | 4.00 |

* Lower value means greater aesthetic preference for a given composition.

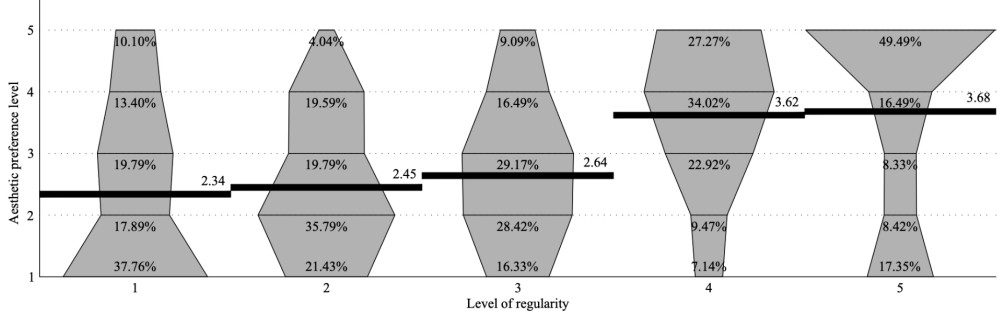

**Figure 7.** Distribution of aesthetic preferences in relation to the degree of regularity of the facade.

It is also interesting to note that 55.65% of the aesthetic preference score for composition no. 1 was above the average, and for composition no. 2 this was 57.22%. Compositions no. 3, 4 and 5 had more than half of the scores below average, and composition no. 5 had 65.85% of these scores. The above results (presented in Figure 8) meant that regular compositions were more often assessed as the most attractive, while irregular compositions had the opposite effect.

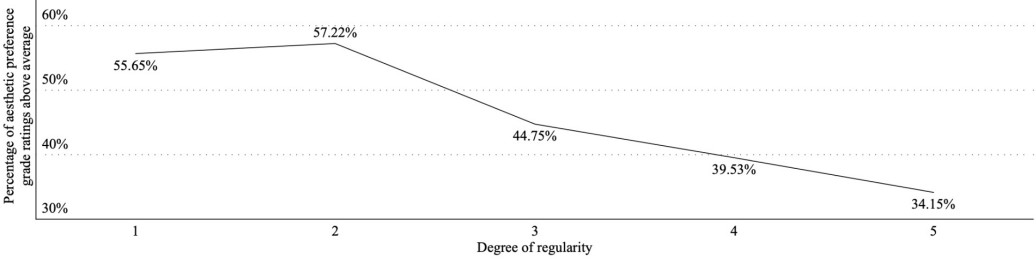

**Figure 8.** Percentage of aesthetic preference grade above avarage.

The conducted analyses indicated a clear relationship between the degree of regularity in the composition and the degree of aesthetic preference. Although the analyses based

on the comparison of means did not prove a linear gradation of correlating variables with each other, the frequency plots showed a different effect. More regular compositions were judged to be more aesthetic than less regular compositions. Equally important is the fact that irregular compositions were assessed as visually unattractive, much more than regular compositions were assessed as attractive. In other words, irregular stimuli elicited a more pronounced effect than regular stimuli.

*3.2. Research Questions 3 and 4*

In order to answer research question three, a one-way ANOVA was performed in an intergroup design. The dependent variable in these calculations was, the type of composition (marked with the letters X, Y and Z), to which the presented stimuli belonged. Dependent variables valued the determining aesthetic preferences in relation to stimuli (on a scale from one, the most attractive, to five, the least attractive). A summary of the results obtained during the above-described operation is presented in Table 4.

**Table 4.** ANOVA summary for the variable "degree of aesthetic preference" depending on the type of composition.

| Source | SS | df | MS | F |
|---|---|---|---|---|
| Between groups | 165.74 | 2 | 82.87 | 45.199 ** |
| Inside groups | 892.88 | 487 | 1.83 | |
| Overall | 1058.62 | 489 | | |

** $p < 0.001$.

The analysis of variance turned out to be statistically significant—F (2.489) = 45.199, $p < 0.001$—, which proved the relationship between the type of composition and aesthetic preferences.

Post hoc tests were performed to see if there were differences in aesthetic preferences depending on the particular type of composition. The result of Levene's test for the variable "aesthetic preferences" turned out to be statistically significant ($p < 0.001$); hence, the assumption of the equality of variance was rejected and Dunnett's T3 test was used. The analyses showed that there was a statistically significant difference ($p = 0.05$) in terms of the aesthetic preferences between composition type X and Z, and also between type B and E. The above results combined with the results of descriptive statistics meant that compositions of type X (M = 2.34; SD = 1.428) and Y (M = 2.55; SD = 1.237) were preferred at a similar level. At the same time, these compositions were more preferred than the Z type compositions (M = 3.65; SD = 1.426). Descriptive statistics are presented in Table 5.

**Table 5.** Descriptive statistics for the dependent variable "Degree of Aesthetic Preference" for composition.

| Function | Type X | Type Y | Type Z |
|---|---|---|---|
| Standard deviation | 1.428 | 1.237 | 1.426 |
| Standard error | 0.144 | 0.088 | 0.102 |
| The minimum value | 2.05 | 2.37 | 3.45 |
| Mean * | 2.34 | 2.55 | 3.65 |
| The maximum value | 2.62 | 2.72 | 3.85 |

* Lower value means greater aesthetic preference for a given composition.

One should also pay attention to the data contained in Figures 9 and 10. The distribution of the frequency of the assessments of the degree of aesthetic preference depending on the type of composition showed that although the means for type X and Y were similar (which resulted from the post hoc tests), the compositions of type X were almost twice as likely to be rated most attractive (37.76% of responses) than compositions of type Y (18.88% of responses). Equally important is the fact that only 36.85% of the respondents assessed

Z-type compositions above average and, what is even more important, only 21.18% of the respondents assessed such compositions as positive (value one or two). The above results clearly indicate that X-type compositions were perceived as more attractive compared to Y-type compositions, and the compositions of both types clearly dominated (in terms of aesthetic preferences) over Z-type compositions.

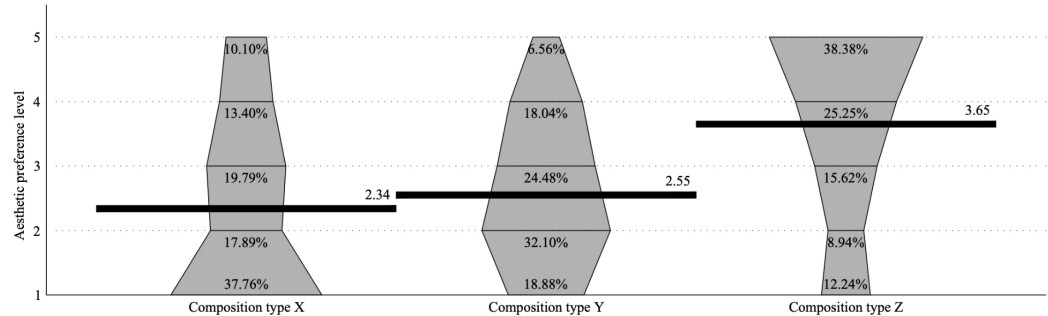

**Figure 9.** Distribution of aesthetic preferences in relation to the composition type of the facade.

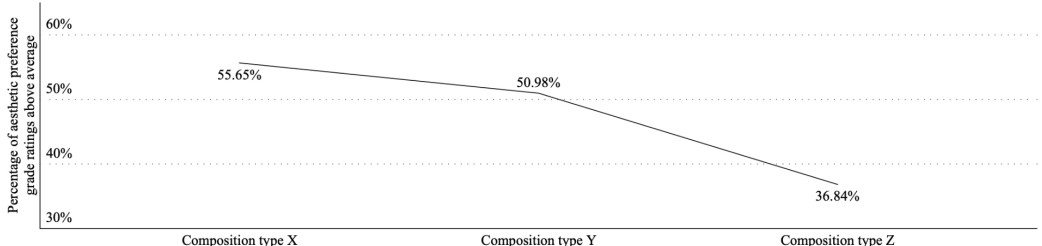

**Figure 10.** Percentage of aesthetic preference grade above average.

The conducted analyses indicated a clear relationship between the type of composition and the degree of aesthetic preference. Statistics based on averages showed that both the X and Y-type compositions were stated to be more attractive than the Z compositions; however, the differences between X and Y types were statistically insignificant. The analysis of the frequency charts showed, however, that also between types X and Y there were clear differences in the degree of aesthetic preference, where X-type compositions were twice as likely to be the most attractive compared to Y-type compositions.

## 4. Discussion

The aim of this study was to determine the relationship between the degree of regularity of the composition and the type of composition of the building facade and aesthetic preferences. Therefore, a questionnaire survey was conducted. Statistical analyses of the collected results created the possibly to answer research questions and formulate conclusions.

The results of statistical analyses clearly indicated the relationship between the degree of the regularity of the composition and the aesthetic preferences for these compositions. Although the correlation index indicated a weak relationship between these two variables, further analyses clearly showed that, generally, compositions with a higher degree of regularity were preferred over the less regular ones. Thus, the compositions of the type X and Y were considered more visually attractive than the compositions of the type Z. It could, therefore, be assumed that the key element for the perception of a given facade composition is the legibility of the vertical composition axes. The compositional axes in the case of Z-type compositions were marked less clearly than the arrangement of elements one above the other according to their axis of symmetry, as in the case of X and Y compositions. It also resulted from the Gestalt principles closeness and continuity [30], according to which we perceive compositional elements one above the other as one vertical line. The grouping of elements vertically and not horizontally results from the vertical nature of architecture [31].

This direction is also suggested by the window openings themselves, which usually have the proportions of a vertically arranged rectangle.

However, a question may be raised as to whether the compositional axes themselves or the clarity of the rule according to which a given composition was created were important in this aspect. Naturally, compositions of type X were subject to the greatest number of constraints and were the most ordered; therefore, the readability of these rules was the greatest in this case. Furthermore, the arrangement of elements one above the other according to the axis of symmetry of each of them was the most natural arrangement, and the compositional axes built by the side edges of the elements (as in the case of Z-type compositions) were not so clear and required a longer analysis. The application of compositional rules, however, was closely related to building compositional axes, although not necessarily in the vertical direction. Therefore, it may not be possible to investigate the correlation between the clarity of a compositional rule and aesthetic preferences in isolating the compositional axes from readability. The results of such analyses would certainly be very interesting and would broaden the knowledge of the perception of composition.

## 5. Conclusions

The results of the study are in line with previous reports on abstract patterns [11–15]. Facades with a regular composition were found more aesthetic than those with an irregular composition. However, the most important for shaping architecture is the effect associated with the negative impact of irregularity, which was clearly stronger than the positive impact of regularity. Such results suggest the need to explore the subject of composition, especially in the context of the perception of irregular compositions (e.g., "Z" type), as it may turn out that they are definitely considered visually unattractive. On the other hand, the conscious use of compositional irregularity may be a desirable visual reinforcement for a dominant or other significant object.

Nevertheless, in the opinion of the authors, formulating general guidelines for designers and architects on the basis of this study would be too hasty. The conducted research should be treated as a starting point for further analyses, deepening the subject of the perception of compositions. First of all, attention should be paid to the problem of the scale of the building. It may turn out that, in the case of larger buildings, regular compositions would not be the most frequently chosen, which would be consistent with the reports of videoecology [16,17]. Relating the research to larger facades would also involve the analysis of compositions consisting of a greater number of elements and would also allow the creation of systems with more subtle differences in the degree of regularity.

Subsequent research should also take into account the influence of other architectural elements on composition preferences. It may turn out that different surroundings, a different colour or material of the façade, different shapes of windows or forms of roofs or the shape of the façade itself would result in a change in preferences regarding the composition.

Another important issue is the multilayer composition theory [2] and the influence of the composition of individual layers on the preference for the composition of the whole. Certainly, the way the stimulus is presented is also important. It would be interesting to see the results of research carried out on the basis of visualizations of buildings from the human perspective. Furthermore, the use of technologies, such as VR, could provide interesting results and would make it possible to arrange a virtual walk; thus, bringing the perception processes occurring during the study closer to those that appear under normal conditions.

This study, in the opinion of the authors, also indicated a very important feature of the composition, which is the degree of regularity. Not only preferences, but also the perception of the degree of regularity in the composition should be the subject of separate research. The results of such analyses could allow for a more conscious design, as they would give designers an understanding of what reactions and perceptions are generated by specific aesthetic treatments.

**Author Contributions:** Conceptualization, M.M. and P.C.; methodology, M.M.; software, M.M.; validation, A.T. and P.C.; formal analysis, M.M. and P.C.; investigation, M.M.; resources, M.M.; data

curation, M.M.; writing—original draft preparation and editing, M.M.; writing—review, A.T. and P.C.; supervision, A.T. and P.C.; project administration, M.M. All authors have read and agreed to the published version of the manuscript.

**Funding:** This research received no external funding.

**Institutional Review Board Statement:** Not applicable.

**Informed Consent Statement:** The consent of the participants was disregarded because the participation in the study was anonymous and voluntary and the questionnaire did not concern sensitive data concerning the participants of the study.

**Data Availability Statement:** The data presented in this study are available on request from the corresponding author.

**Conflicts of Interest:** The authors declare no conflict of interest.

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
