# Peer review of "Preferences of the Facade Composition in the Context of Its Regularity and Irregularity"

_buildings, doi:10.3390/buildings12020169_

Round 1

Reviewer 1 Report

The article is sound and precise. Simple and easy to understand. 

It is impossible to understand the values standing behind the decision to focus exclusively on the articulation of the plain building view. It might be the most important of all, but not the only determinant that shapes ones opinion and acceptance of architecture. 

In my opinion, the complexity of perception i.e. understanding and accepting the visual identity of architecture is way more layered then the article shows. Therefore, I suggest that instead of only imagining the articulation of an abstract front view of the imaginary building, the author should make a second round of investigation using the abstracted front view of the existing Buildings shown in the introductory part of the article. One may say that regularity or irregularity on paper is one thing but in real life it is absolutely a different matter. Then, one could compare the answers and see what other criteria are important when appreciating architecture is in question. 

Therefore, it would be good to further the research, or to add an additional discussion part opting for further possible research topics to be included in future research, with elaborate explanation and possible goals.  

Author Response

Thank you for your time and proofreading guidelines.

Please see the attachment. Referring to the received review, the text has been corrected in terms of describing the desirability of reducing architecture to a two-dimensional façade [Line 138-140]. The chapter "Conclusions" was also extended to include a proposal for further research directions and the observed weaknesses of the study [Line 334-336].

Reviewer 2 Report

The submitted article presents a relevant issue to the field of Architecture, observing the acceptance of new architectural languages and forms by public opinion and, it can be a significant contribution to scrutinizing the success of some proposals that have strong support from the media.

The objective of the study was to understand the Poles “aesthetic preferences” [line 98] taking into account the geometrical composition of building facades. It is proposed that the sample of the survey carried out may represent either the taste of Polish society or, later, a universal tendency [line 37]. Overall, the methodology used is adequate to the proposed objectives.

Reading the paper may lead to some misunderstandings which, in our view, require a review, namely the introduction (chapter 2), the discussion (chapter 5) and the conclusion (chapter 6). The first problem concerns the application of the term “Aesthetics” in architecture (i); the second, is relative to doubt of taking the part (the constructive element “facade”) to refer to the entirety (ii); the third, it's about the possible mistake of taking the small universe of the inquiry as representative of a universal taste (iii).

(i) The title - “Aesthetic preferences in the context of the regularity and irregularity of the facade composition” – highlights the aesthetic sense of the population when facing several geometric composition of  voids in the plan of the main facade. The term “aesthetics” in architecture is covered with complex properties that go far beyond the simple visual perception of the qualities of a vertical plan as it is presented in the study. See, for example, the meaning given by Roger Scruton in “The Aesthetics of Architecture”. Thus, it is suggested either the replacement of the term, or a more in-depth explanation of the use of the word in the text, perhaps relying on other specialist authors to do the contradictory. This explanation can be given in the introduction.

(ii) The study takes the building for one of its constituents - the main facade - removing, among other aspects, the three-dimensional character, the surroundings context, or its material condition like textures, shadows, construction materials, color, etc. The perception, even reduced to its visual character, is dynamic and supposes an apprehension of the form that is informed by multiple sensorial and cultural aspects. It's not observed in the paper the reason why such a drastic reduction is made about the complexity of the observer/object perception phenomena. Rudolf Arnheim in the quoted book [reference 17] - The Dynamics of Architectural Form – raises doubts about the possibility of highlighting parts of the object destroying the idea of ​​the whole (p.4 - Introduction), however it opens the hypothesis of this being done because it is operative to certain analysis conditions. It is extremely important explain how valid it is this position to verify the sympathy or antipathy for certain facade compositional rules. And how it's possible from here establish a trend of popular taste for contemporary architectural movements. This explanation can be given in the introduction.

(iii) In the context of the dimension and composition of the Polish population, the universe of analysis (109 responses) seems to be very small and it's not be given qualitative indicators about the participants, namely, in such aspects that can denounce some prejudices: rural or urban inhabitants; and jobs. Thus, it must be explained how the survey sample is representative of the Polish population. It will be difficult to extrapolate to other societies than Poland, with other cultural references, the same perception. I suggested that this explanation can be given in the context of the discussion.

Such experience limitation that served to answer the survey - erasing  a broader understanding of the building, or limiting the diversity of facade to only five variations of the same design composition (for example, what would be the result if circular windows were used even on a regular and symmetrical layout) - weakens the argument necessary to support the final statement: “Facades with a regular composition have been found more aesthetic than those with an irregular composition” [line 298]. It is recommended that point 6 [Conclusion] be developed further, assuming the partial relevance of the study as a contribution clarifying its limitations and circumstances.

Regarding the structure of the contents, note that chapter 1 maintains the original text of the template [line 16 - 22].

Author Response

Thank you for your time and proofreading guidelines.

    1. The title was corrected by removing the word „aesthetic". The authors decided that it is not important to convey the actual topic of the work.
    2. The text has been corrected in terms of describing the desirability of reducing architecture to a two-dimensional façade [Line 138-140].
    3. The substantive basis for determining the size of the research sample was described, and statistical calculations were made to determine the margin of error [Line 188-190].
    4. The substantive basis for limiting the number of stimuli presented in the study to five was described [Line 126-131].
    5. The chapter "Conclusions" was extended to include a proposal for further research directions and the observed weaknesses of the study [Line 334-336].
    6. The original text of the template has been removed.

Reviewer 3 Report

Regularity is just one element of aesthetics. Given that aesthetic considerations in building design are inherently multi-attribute (as illustrated in Figure 1), this line of inquiry is likely to advance to address preferences among designs viewed as packages of attributes in a variety of combinations. Techniques for studying preferences in this broader context are well-developed and employed in many related contexts. Choice experiments, for example, have the advantage of evaluating aesthetic attributes in a multi-attribute context that evaluates trade-offs, includes cost considerations, and produces estimates of contribution to value.

A few specific observations:

Line 12. “The above-described discoveries shed, in the authors’ opinion, a completely new light on the contemporary work of architects and suggest a necessity of revisioning the current trends.” It is not clear that architects should defer to ordinary citizens in matters of aesthetics. Rather, I envision a continuing (metaphorical) conversation in which architects challenge citizen aesthetic expectations and citizens’ views evolve but within limits. Architecture is, among other things, an art and should not be obligated to give people only those aesthetic attributes that they already prefer.

Lines 70-87. This is really provocative material for those of us who are not familiar with it. Visual monotony is associated with loss of brain cells. Beauty stimulates the brain’s reward system. But these ideas don’t seem to have had much influence on the authors’ conclusion that architects should rein-in their deviations from regularity.

Fig 3 highlights the uni-dimensional framing of this work. Without more visual context the irregular placement of windows seems pointless, but I can imagine more favorable preferences for a façade with additional visual cues that provide a context for it.  

Author Response

Thank you for your time and proofreading guidelines.

    1. The abstract has been corrected. The idea of finding the golden mean between the utility of architecture and meeting users' expectations was supplemented with the necessity to implement paradigms related to the nature of architecture as art [Line 13-15]. 
    2. The idea of videoecology discoveries was developed in the context of the need to study preferences for facade composition [Line 68-74].
    3. The text has been corrected in terms of describing the desirability of reducing architecture to a two-dimensional façade [Line 138-140]. 

Round 2

Reviewer 2 Report

The text now presented is more balanced in terms of the problems posed before. However, the term "aesthetics" continues to be used in the text without prior concern to define it precisely.
We do not agree with what is written in lines 330 and 331 since it is not possible to take a survey that is partial to support an indication of the architecture's way of doing which is circumstantial to the material, cultural and political conditions of a geographical space.

Author Response

Thank you for your time and proofreading guidelines.

Please see the attachment. Referring to the received review, the text has been corrected in following points:

[Lines 5-6] Reformulated „aesthetic preferences” to „preferences towards visual attributes of objects”

[Line 11] Word „aesthetic” changed to „attractive”.

[Line 14] Word „aesthetics” changed to „visual dimension.

[Lines 22-28] Explanation of the meaning of the word "aesthetics" used in the context of this study.

[Lines 39-54] A paragraph on the concept of aesthetics has been added

[Lines 130-141] Explains the purposefulness of insulating the composition of window openings as a component of the building, affecting the aesthetics of its entirety.

[Lines 354-358] The authors of the study agree with the reviewers. The conclusions regarding irregular compositions have been reformulated into more general ones.
